

# Carbon cycling at the aquatic-terrestrial interface is linked to parafluvial hyporheic zone inundation history

Amy E. Goldman[1], Emily B. Graham[1], Alex R. Crump[1], David W. Kennedy[1], Elvira B. Romero[1], Carolyn G. Anderson[1], Karl L. Dana[1], Charles T. Resch[1], Jim K. Fredrickson[1], and James C. Stegen[1]

[1]Pacific Northwest National Laboratory, Richland, 99352, USA

*Correspondence to*: Amy E. Goldman (amy.goldman@pnnl.gov)

**Abstract.** The parafluvial hyporheic zone combines the heightened biogeochemical and microbial interactions indicative of a hyporheic region with direct atmospheric/terrestrial inputs and the effects of wet/dry cycles. Therefore, understanding biogeochemical cycling and microbial interactions in this ecotone is fundamental to understanding carbon cycling at the
aquatic–terrestrial interface and to creating robust hydrobiogeochemical models. We aimed to (*i*) characterize biogeochemical and microbial differences in the parafluvial hyporheic zone across a small spatial domain (6 lateral meters) that spans a breadth of inundation histories and (*ii*) examine how parafluvial hyporheic sediments respond to laboratory-simulated reinundation. Surface sediment for assays and forced inundation laboratory incubations (destructively sampled at 0.5 hours and 25 hours) was collected at four elevations along transects perpendicular to flow of the Columbia River, eastern
WA, USA. The sampling elevations were inundated by the river 0 days, 13 days, 127 days, and 398 days prior to sampling. Spatial variation in environmental variables (organic matter, moisture, nitrate, glucose, % C, % N) and microbial communities (16S and ITS rRNA gene sequencing, qPCR) were driven by differences in elevation and thus inundation history. Microbial respiration did not differ significantly across elevations prior to inundation. Inundation suppressed microbial respiration relative to uninundated sediment across all elevations, but the degree of suppression was dramatically
different between the elevations saturated and unsaturated during sampling, indicating a binary threshold response. We present a conceptual model in which irregular hydrologic fluctuations facilitate microbial communities adapted to local conditions and a relatively high flux of $CO_2$. Upon re–wetting, microbial communities are initially suppressed metabolically, which results in lower $CO_2$ flux rates primarily due to suppression of fungal respiration. Following prolonged inundation, the microbial community adapts via a shift in composition. Our results indicate that the time between inundation events can push
the system into alternate states: we suggest that (*i*) above some threshold of inundation–interval, re–inundation suppresses respiration to a consistent, low rate, and (*ii*) that below some inundation–interval, re–inundation has a minor effect on respiration. Extending reactive transport models to capture processes that govern such dynamics will provide more robust predictions of river corridor biogeochemical function under altered surface water flow regimes in both managed and natural watersheds.





# 1 Introduction

The hyporheic zone, the subsurface region where river water and groundwater (GW) mix, is characterized by heightened biogeochemical cycling and microbial interactions (McClain et al., 2003; Boulton et al., 2010; Mulholland and Webster, 2010; Stegen et al., 2016). Hyporheic sediment is exposed to different aquatic chemistries due to variation in hydraulic head that drives flow into and out of the hyporheic zone (Arntzen et al., 2006; Hucks Sawyer et al., 2009). When river elevation rises, river water infiltrates hyporheic sediment. When river elevation falls, outflow from the hyporheic zone consists of both bank storage, which contains river water altered by hyporheic biogeochemical transformation (Wörman et al., 2002), and groundwater flowing out to the river through the hyporheic zone (Boano et al., 2008). Many prior studies have focused on the main channel hyporheic zone that is continuously overlain by a column of water (e.g., Boulton et al., 1998; Tonina and Buffington, 2009; Boulton et al., 2010; Mulholland and Webster, 2010; Trimmer et al., 2012). However, the parafluvial hyporheic zone (Fig. 1), located in the region of the river channel that is dry during low flows, presents an important location to investigate biogeochemical cycling at the aquatic-terrestrial interface because it combines the prominent interactions of a hyporheic region with the addition of direct atmospheric/terrestrial inputs and the effects of wet/dry cycles. Given that the parafluvial zone is present in all river systems, understanding biogeochemical cycling and microbial interactions in this ecotone is fundamental to understanding the coupling between aquatic and terrestrial carbon cycles.

In dam-controlled watersheds, the parafluvial hyporheic zone undergoes rapid cycles of wetting and drying caused by hydropeaking regimes in which flow is intentionally maximized during times of high energy demand and/or fish migration. In the Hanford Reach of the Columbia River, for example, hydropower dam operations cause river water elevation to fluctuate up to 2 meters in a single day (Arntzen et al., 2006). Over time, cycles of wetting and drying impact different elevations of the parafluvial zone at different frequencies, which naturally results in a gradient of sediment moisture content, but also has the potential to create biogeochemical and microbial interactions specifically dependent on preceding environmental conditions. By identifying the processes impacted by the historical contingencies of this variable inundation, it is possible to understand linkages among hydrology, biogeochemistry, and microbial ecology on a systems level, which therefore allows the creation of robust hydrobiogeochemical models.

Previous studies of biogeochemical cycling in the parafluvial zone have sampled gravel bars to relate flow paths and water residence time to biogeochemistry and microbial activity (Claret and Boulton, 2008; Deforet et al., 2009; Zarnetske et al., 2011) and have sampled during saturated conditions utilizing well/piezometer transects (Baker et al., 1999; Deforet et al., 2009; Briody et al., 2016; Graham et al., 2016; Stegen et al., 2016). Several studies have also investigated how biogeochemical cycling in the parafluvial zone compares to other floodplain environments (e.g., forest, ponds, islands, wetlands) (Burns and Ryder, 2001; Doering et al., 2011; Ostojić et al., 2013; Bodmer et al., 2016). Additionally, Zeglin et al. (2011) investigated parafluvial transects of two intermittent streams located in cold (Antarctic) and hot (New Mexico, US)





deserts and identified binary microbial community (16S rRNA gene sequencing) differences based on wet vs. dry sediments. With regard to wet/dry cycles, many studies have examined their biogeochemical and microbial effects in terrestrial soils (Jarvis et al., 2007 and references therein;Kim et al., 2012 and references therein). Fewer have investigated them in intermittent stream sediments, wherein the entire stream dries for stretches of time (Leigh et al., 2016 and references

therein). Prior work has not examined how inundation history and wet/dry cycles influence biogeochemical cycling and microbial communities within a hydropower-influenced transect of the parafluvial hyporheic zone and has not investigated the degree to which response to rewetting is due to inundation–history–driven microbial community composition or aqueous chemistry. Determining drivers of biogeochemical– microbial interactions in the parafluvial zone will elucidate the implications of climate-watershed interactions on carbon cycling, which will aid understanding of both managed river

systems and related ecosystems (e.g., intermittent streams, tidally-influenced littoral zones of the Great Lakes).

To fill this knowledge gap in parafluvial biogeochemical cycling dynamics, we aimed to (*i*) characterize differences in sediment biogeochemical and microbial (bacteria, archaea, and fungi) variables in the parafluvial hyporheic zone across a spatial domain of 6 lateral meters that integrates a breadth of inundation histories and (*ii*) examine how the sediments

respond to laboratory–simulated re–inundation. In order to determine the re–inundation response under different GW/river water mixing scenarios, we used three different re–inundation waters to capture the influence of cation content (typically higher in GW), $NO_3^-$ concentration (typically higher in GW), and availability of dissolved organic carbon (typically higher in river water).  We expected to observe higher amounts of carbon (e.g., organic matter, glucose) at the upper transect elevations where it could accumulate away from the physical mixing and dilution of inundation and subsequently greatest

respiration at the drier upper elevations following re–wetting (e.g., the "Birch Effect"; Birch, 1958).  We identified two opposing hypotheses to determine additional drivers of microbial respiration across an elevational transect of the parafluvial hyporheic zone. First, we hypothesized that recreating the GW-river water mixing that occurs in the hyporheic zone would stimulate $CO_2$ flux regardless of inundation history, because GW-surface water mixing brings complementary electron donors and acceptors together (McClain et al., 2003). Alternately, we hypothesized that inundation history is a more

important driver of respiration response, such that local adaptation of the microbial community due to inundation history would overwhelm the influence of shifts in aqueous chemistry.

## 2 Materials and Methods

### 2.1 Study site

Samples were collected on 28 March 2016 from the parafluvial zone of the Columbia River located 80 km downstream of

hydroelectric Priest Rapids Dam in the 300 Area of the Hanford Reach in semi-arid eastern WA, USA (Arntzen et al., 2006; Slater et al., 2010). Precipitation in the two weeks prior to sampling included trace amounts the week before sampling and cumulative rainfall of 0.64 cm from March 13 to 22 (Hanford Meteorological Station, 2016). River water elevation in the





Hanford Reach is controlled by dam operations and fluctuates up to 2 meters daily (Arntzen et al., 2006). A water column pressure sensor (0–15 psig; Campbell Scientific) was used to record water level elevation near the study site at 15–minute intervals from 2013 to 2017. In order to relate shoreline elevation to historical river water elevation, an elevation survey of the study site was performed by combining manual elevation measurements with a point of known elevation for each sampling transect, which allowed for the calculation of time since last inundation by the river for each sampling elevation.

## 2.2 Experimental design

Surface sediment (0–10 cm) was collected at four locations along three transects (12 sites total) perpendicular to river flow and encompassing 1 m of elevation change (Fig. 1). The four locations were spaced laterally at 2 m intervals beginning at the water line and going 6 lateral meters upslope. The three transects were within 4 m of each other. For clarity, the four elevations will be referred to by their distance from the water line: 0 m, 2 m, 4 m, and 6 m. The highest elevations (4 m and 6m) were marked by grasses and mature trees (*Morus rubra*; Red Mulberry) with large roots extending down through the 4 m elevation. The lower elevations (0 m and 2 m) were marked with sparse grasses and dried algal mats. All elevations had an extensive cobble layer overlying the sediment. Based on recorded river water elevation, the four sampling elevations had been last inundated by the river 0 days (0 m sampling location), 13 days (2 m), 127 days (4 m), and 398 days (6 m) prior to sampling. Unsieved sediment was collected at each location for grain size analysis and for organic matter (OM) content (loss-on-ignition; LOI). For all remaining analyses, sediment was sieved to < 2 mm and subsampled in the field. 15 mL aliquots for laboratory incubations were subsampled into 40 mL borosilicate incubation vials and kept on ice until used for same–day laboratory incubations. Aliquots for microbial analyses were flash frozen in liquid $N_2$ and kept on dry ice before storage at −80°C. Aliquots for C/N (Elementar vario EL Cube), sieved OM content (LOI), and moisture content (gravimetric) were stored on dry ice before storage at –4°C.

## 2.3 Laboratory incubations

Incubation vials were removed from ice in pairs to be used for destructive sampling at 0.5 hour and 25 hour time points. Vials were left at room temperature for 10 minutes before 15 mL of re–inundation treatment water was added to each vial, leaving a 10 mL headspace of ambient air. Three replicate vials from each sample site were used per time point and were exposed to one of three different treatment waters: river water (collected during sediment sampling; stored on ice), 1:1 river water and synthetic GW without nitrate ($NO_3^-$), or 1:1 river water and synthetic GW with $NO_3^-$. Both synthetic GW solutions were created from calcium chloride, sodium bicarbonate, magnesium sulfate, sodium sulfate, sodium carbonate, potassium carbonate, and (for $NO_3^-$ containing GW) calcium nitrate. The final content of each GW contained (mmol/L): 1.4 $Na^+$, 1.0 $Ca^{2+}$, 0.5 $Mg^{2+}$, 0.2 $K^+$, 1.6 $CO_3^{2-}$, 0.6 $SO_4^{2-}$. The $NO_3^-$ containing GW had 0.5 $NO_3^-$ and 1.3 $Cl^-$, whereas the non-$NO_3^-$ GW had 1.8 $Cl^-$. The final pH values of the three treatments were 7.59 (river water), 7.77 (River + GW), and 7.66 (River + GW + $NO_3^-$). Vials were inverted multiple times and then stored upright (dark, room temperature) until their end–time sampling point.




After the incubation period, vials were shaken on a mini-vortex for three minutes to equilibrate dissolved gas with headspace. Vial headspace was subsequently sampled using a 5mL gas–tight syringe and immediately analyzed for $CO_2$ (EGM–4, PP Systems, Amesbury, MA). $CO_2$ values (ppm) were used as a proxy for respiration rate (RR; ppm $CO_2$ min$^{-1}$ g

dry sediment$^{-1}$) after initial (0.5 hr) and prolonged (25 hr) inundation. After headspace sampling, vials were opened and subsampled. Assays on water-extracted samples included anions (Dionex ICS–2000 anion chromatograph with AS40 auto sampler), cations (nitric acid acidified; Perkin Elmer Optima 2100 DV ICP–OES with an AS93 auto sampler), pH, and dissolved non-purgeable organic carbon (NPOC; Shimadzu combustion carbon analyzer TOC–Vcsh with ASI–V auto sampler). Ammonium ($NH_4^+$) was also determined colorimetrically, following KCl extraction (adapted from Weatherburn,

1967) (Shimadzu UV/Vis spectrophotometer).

In addition to incubation vials that had treatment water added, a set of untreated, control incubations (one from each site) was incubated for 0.5 hours without the addition of any water, and vials were subsequently analyzed for organic acids and sugars (Agilent 1100 series HPLC), headspace $CO_2$, cations, and anions as described above.

### 2.4 Microbial analyses

DNA was extracted from flash-frozen sediment using a MoBio PowerSoil kit (MoBio Laboratories, Inc., Carlsbad, CA) according to manufacturer's instructions with the addition of one-hour incubation with proteinase-K solution (Applied Biosystems, Foster City, CA) and MoBio C1 solution to facilitate cell lysis. 16S and ITS rRNA amplicon sequencing was

used to identify microbial and fungal communities. Assays were performed in triplicate using an Illumina MiSeq at the Environmental Sample Preparation and Sequencing Facility at Argonne National Laboratory, Lemont, IL. The recently modified forward barcoded primer set was used for 16S sequencing (Apprill et al., 2015), and reverse barcoded primers were used for ITS sequencing.

Quantitative PCR (qPCR) was used to quantify 16S and ITS rRNA gene copies. Assays were performed in 384-well plates using a Life Technologies ViiA7 real-time PCR instrument at the DNA Services Facility at the University of Illinois at Chicago, using methods described previously (Nadkami et al., 2002). Final gene copies per extract were calculated by multiplying elution volume by gene abundance and were then normalized to grams of dry sediment extracted. Absolute concentrations (referred to as "abundance" throughout) of specific OTUs were calculated by multiplying percent abundance

from 16S and ITS sequencing by the sample's total bacterial, archaeal, or fungal concentration (ng ul$^{-1}$ g dry sed.$^{-1}$) from qPCR.



## 2.5 Statistical analyses

All statistical analyses were completed using R (version 3.2.4). Linear and quadratic regressions were used to assess the relationship between environmental variables and elevation. If the variable clearly exhibited a stepped relationship, analysis of variance (ANOVA) and subsequent pairwise t–tests were used to categorically compare the elevations to each other. Using days since inundation instead of distance from water line did not qualitatively change the results or any conceptual inferences. We used distance because it is a more integrative measure of inundation history. $R^2$ values reported are adjusted $R^2$. Significance for all analyses was determined based on $\alpha = 0.05$. Operational taxonomic unit (OTU) data were rarefied for both 16S and ITS sequencing based on the sample with the lowest count of assigned OTUs, because assigned 16S OTU counts ranged from 44,807 to 345,564. Rarefied data were used for all subsequent analyses, including calculation of absolute concentrations utilizing qPCR results. The *Chloroplast* class was removed from 16S data analyses, because it is often a sequencing artifact and not indicative of sample composition. Permutational multivariate analysis of variance (PERMANOVA) and non-metric multidimensional scaling (NMDS) were used to assess dissimilarities among bacterial/archaeal and fungal communities using Bray-Curtis distances calculated from 16S and ITS OTUs within the *vegan* R package. Significant environmental variables were plotted as vectors on NMDS plots using the "envfit" function from *vegan*. Because NPOC was not measured in the untreated samples, NPOC at 0.5 hours was averaged for each sampling site to be used for statistical analyses. Linear discriminant analysis coupled with effect size (LEfSe) was performed following Niederdorfer et al. (2016). LEfSe utilizes Kruskal-Wallis sum rank tests ($\alpha = 0.05$), pairwise Wilcoxon rank-sum tests ($\alpha = 0.05$), and linear discriminant analysis (threshold= 2.0; strategy="all–against–all") to detect differential taxa among the four elevations and estimate effect size. Analyses were completed and figures were produced using the Huttenhower Galaxy server (http://huttenhower.sph.harvard.edu/galaxy).

## 3 Results

### 3.1 Field conditions

#### 3.1.1 Spatial gradients in environmental variables

Moisture content of the sediment decreased with increasing elevation, with mean (sd) percent moisture of 35.5 (6.6), 24.5 (2.2), 21.5 (1.7), and 17.5 (1.8) and mean percent field saturation 100, 69, 61, and 49, from 0 m, 2 m, 4 m, and 6 m, respectively. In addition, many environmental variables (organic matter (<2mm sieved), glucose, $NO_3^-$, % C, and % N) exhibited significant positive linear relationships with elevation (Table 1), but some variables had more complex spatial distributions linked to elevation (Fig. 2). Specifically, organic matter displayed a distinct stepped distribution in which 0 m was significantly lower than the upper elevations (2 m, 4 m, and 6 m) (ANOVA and pairwise t–test; P<0.01 for all), and the upper elevations did not differ from each other (P >0.7 for all). In addition, glucose and % C also exhibited stepped



distributions, but in both cases, 0 m was significantly lower than only 4 m and 6 m (% C: P <0.01 for 4 m and 6 m; glucose: P <0.04 for 4 m and 6 m) (Fig 2).

### 3.1.2 Microbial biogeography

Fungal to bacterial ratio and fungal abundance were significantly positively related to elevation (Table 1). Conversely, bacterial abundance lacked a linear relationship with elevation but displayed a distinct stepped distribution in which 0 m was significantly lower than the upper elevations (2 m, 4 m, and 6 m) (ANOVA and pairwise t-test; P <0.03 for all), but the upper elevations were not significantly different from each other (P >0.1 for all), much like the stepped distribution between distance and organic matter, glucose, and % C (Fig. 2).

From 16S and ITS sequencing, 9893 and 1856 individual OTUs were identified, respectively (Table S1). The same seven fungal phyla were identified at all elevations from ITS sequencing. Although the order of abundance varied across the individual sites, all four elevations had the same top eleven bacterial and archaeal phyla identified from 16S sequencing (Table S2). Despite the similarities on the phylum-level, the bacterial/archaeal and the fungal community compositions were significantly different across elevations based on OTUs (PERMANOVA, bacteria/archaea: P<0.001; fungi: P<0.01) (Fig. 3). Of 153 bacteria/archaea classes identified by 16S sequencing, 15 were unique to the 0 m elevation, and two were present at all elevations but absent at 0 m. Of the 15 unique classes, the most abundant were Marine Group I and South African Gold Mine Gp 1, which are both in the ammonia oxidizing *Thaumarchaeota* phylum (Francis et al., 2005; Brochier-Armanet et al., 2008). Additionally, LEfSe identified no archaeal, no fungal, and 12 bacterial taxa driving dissimilarity among the elevations (Fig. S1). All of the LEfSe–identified taxa were from the 0 m and 6 m elevations.

Bacterial/archaeal and fungal communities clustered in distinct elevation groups in NMDS plots (stress= 0.01 and 0.04, respectively; Fig. 3a and b). Significant environmental variables plotted as vectors on NMDS plots indicate moisture content was the strongest predictor of both bacterial/archaeal and fungal communities, with 0 m clustering separately from the upper elevations and $R^2$ for moisture of 0.90 and 0.91, respectively. Additionally, carbon was a significant explanatory variable in both communities in multiple forms (e.g., % C, OM, glucose, NPOC).

## 3.2 Incubations

### 3.2.1 Response to different treatment waters

The initial (0.5 hour), prolonged (25 hour), or change (25hr–0.5hr) in any measured inundation response did not differ significantly across treatment waters (river water; 1:1 river water and synthetic GW without $NO_3^-$; 1:1 river water and synthetic GW with $NO_3^-$) (ANOVA: P>0.05), despite large differences in $NO_3^-$ between treatments and source sediment



(Table 2). Given the lack of treatment effect, the results from the three treatments were pooled and utilized as replicates for all other incubation-based results.

### 3.2.2 Respiration response to inundation

In control sediment (no water added), respiration rate (RR) across all elevations was not significantly different (ANOVA; P>0.2). Although initial and prolonged inundation suppressed RR relative to control sediment across all elevations, the degree of suppression was dramatically different between 0 m and the upper elevations (Fig. 4). Specifically, following experimental inundation, the samples saturated at time of sampling (0 m) had a significantly higher RR than those unsaturated at time of sampling (2 m, 4 m, 6 m) (ANOVA and pairwise t–test; P <0.001), and the unsaturated elevations did not have RRs significantly different from each other (pairwise t–test; P >0.07) (Fig. 4b,c). Across the elevations, therefore, the RR response to inundation was binary, with greater suppression relative to uninundated rates at the upper elevations and the highest RR at 0 m.

## 4 Discussion

### 4.1 Microbial respiration response to inundation

The RR suppression at the upper elevations relative to 0 m after both initial and prolonged inundation (Fig. 4a,b,c) indicates that brief increases in moisture content (e.g., recurring splashing or < 1 day periods of hydropeaking) can suppress respiration rates in the parafluvial hyporheic zone. Numerous studies have reported high $CO_2$ production when dry soils are wetted (e.g., the "Birch Effect"; Birch, 1958;Fierer and Schimel, 2003) and greater $CO_2$ from soils undergoing wet/dry cycles than those maintained at a specific moisture content (Miller et al., 2005). We therefore had hypothesized that high amounts of carbon (% C, glucose, NPOC, OM) and bacterial abundance would fuel respiration at the unsaturated upper elevations (2 m, 4 m, 6 m) immediately following inundation, despite the upper elevations being at 49–69% field saturation; however, we found that prior to inundation, RR did not differ significantly across elevations, but after inundation, the upper elevations had significantly lower RR than 0 m. Consistent RR across the upper elevations was unexpected, because they differed significantly in features that influence RR, including microbial composition, surrounding vegetation, and time since most recent river inundation. We suggest that microbial communities in sediments that are inundated permanently or for extended periods are better adapted to full water saturation than those that are infrequently inundated, which led to the binary RR response between 0 m and the upper elevations.

The RR suppression within the upper elevations following re-inundation could be attributed to multiple factors. First, the upper elevation microbial communities may have physiologically adapted to prolonged terrestrial conditions, and the sudden return to aquatic conditions may have led to osmotic shock or other stress response that produced a lower RR than that of the already saturated samples and contributed to a lag in response time (Kieft et al., 1987; Fierer et al., 2003; Jarvis et al., 2007;





Schimel et al., 2007; Borken and Matzner, 2009; Warren, 2014; Meisner et al., 2015). Second, as suggested above, the upper elevation microbial community composition may have been less suited to respiring carbon in aquatic conditions or during wet/dry cycles due to the higher proportion of fungi (Lundquist et al., 1999). Fungi have been shown to account for the majority of soil/sediment respiration, with variability across studies yielding estimates of respiration contribution of 78%

5    (Anderson and Domsch, 1973), 74–90% (Blagodatskaya and Anderson, 1998), and 18–40% (Sapronov and Kuzyakov, 2007). As filamentous fungi are poorly adapted to soils/sediments that are water-saturated or inundated for extended periods (Alexander, 1977; Kominkova et al., 2000; Kuehn et al., 2000), it follows that for upper elevation sediment that experiences less frequent and shorter intervals of inundation, microbial respiration will be depressed upon re-inundation due to fungal suppression.

Both scenarios may have contributed to the RR response to inundation, but an osmolytic microbial stress response was unlikely considering the relatively high moisture content of the sediments prior to inundation and the rainfall that had occurred in the weeks prior to sampling. Given the expected fungal response to inundation and the higher proportion of fungi at the upper elevations, it is most likely that community composition–in particular the distribution of fungi–was a key driver

to the RR response. The persistence of the RR spatial distribution after prolonged inundation further suggests a significant time lag in the response of the upper elevations following perturbation, which can be attributed to the need for community shifts in response to inundation.

Within the context of the elevation-based responses to re-inundation, the absence of re–inundation response differences

among the three treatment waters suggests that inundation history is a stronger driver of short–term biogeochemical dynamics than shifts in aquatic chemistry. Large fluctuations in river water elevation alter hydraulic flow paths at the study site (Arntzen et al., 2006) and subsequently determine whether surface sediments are exposed to river water, bank storage of river water altered by hyporheic biogeochemical transformation (Wörman et al., 2002), or GW flowing out to the river through the hyporheic zone (Boano et al., 2008). The treatment waters were intended to reflect the range of aqueous

chemistry conditions resulting from the exposure to these different water sources. The lack of difference among the treatment waters was unexpected, particularly for $NO_3^-$. In the upper elevations where $NO_3^-$ was highest (Table 2), the sediment values likely overwhelmed the influence of the $NO_3^-$ treatment, but this should not have been the case for the lower elevations, where we expected reduction of $NO_3^-$ and potentially associated production of $NO_2^-$ or $NH_4^+$. Locations with longer times-since-inundation accumulated more biogeochemically reactive water soluble analytes (e.g., $NO_3^-$, glucose) that

would not have suppressed RR upon re-inundation; if anything, we would expect them to stimulate RR. This, combined with lack of response to experimental treatments, strongly suggests that the influence of inundation history is via the microbial community. In this case, non-inundated conditions select for a suite of traits that allow for growth and reproduction under relatively dry to moist conditions, with the apparent tradeoff of metabolic suppression upon re-inundation. As such, we propose that inundation history influences short-term biogeochemical responses to re-inundation by imposed ecological



selection for traits that fall along a tradeoff surface that spans fungal and bacterial/archaeal strategies. This is similar to Chambers et al. (2016), who reported greater impacts on microbial community structure and function from inundation history relative to water chemistry (salinity) in mangrove peat soil.

## 4.2 Microbial communities differ along inundation history gradient

Examining patterns of microbial community structure and relative abundances of fungi and bacteria along the elevation transects provides evidence in support of inundation history influencing biogeochemical responses via selection for locally adapted microbial communities. At 0 m, the fungal to bacterial ratio and the abundance of fungi were lower than those at the upper elevations that experienced longer periods of unsaturated conditions. Other studies have identified similar patterns in which the ratio of fungal to bacterial biomass, based upon phospholipid fatty acid (PLFA) biomarkers, was consistently

lower in flooded vs. unsaturated soil (Bossio et al., 1998), and the fungal biomarker 18:2ω6,9c was absent in in long-term submerged floodplain soil relative to floodplain soils experiencing shorter inundation time (Rinklebe and Langer, 2006). With regard to bacteria, LEfSe only identified bacterial taxa driving dissimilarity from 0 m and 6 m elevations (Fig. S1), which suggests that there was a gradient in bacterial community composition in which the two inundation history end-members paired with community end-members. Therefore, inundation history likely drove the spatial distribution of

microbial community composition, which supports the inference that the response to re-inundation was influenced by the specific community make–up (e.g., proportion of fungi).

## 4.3 Importance of parafluvial hyporheic dynamics to hydrobiogeochemical models

Biogeochemical response lags due to physiological stress of parafluvial microbial communities are of particular importance in creating modeling frameworks that can accurately predict the biogeochemical behavior of terrestrial-aquatic transition

zones in managed systems. Microbial adaptation to the abrupt change in local environment is not instantaneous in general, but a long time lag is often observed (Wood et al., 1995). This phenomenon can affect the transport of biodegradable substrates and contaminants, particularly when the timescale of the metabolic lag is comparable to that of transport (Nilsen et al., 2012). Despite such importance, mechanistic approaches for incorporating metabolic lags into subsurface transport modeling are currently lacking. Past modeling work that attempted to account for this lag effect was either based on temporal

convolution integral (Wood et al., 1995; Nilsen et al., 2012) or introducing exposure time as an additional dimension (or coordinate) (Ginn, 1999). These efforts do not address fundamental mechanisms responsible for time lags and consequently, their predictions may be limited.

Our results indicate that a combination of ecological and physiological mechanisms impeded the ability of the microbial

communities to rapidly adapt to an inundated state. This resulted in the suppression of RR, which we propose is a transient condition whereby there is also temporal lag in biogeochemical function. Therefore, future efforts need to consider more




ecologically and physiologically relevant causes for delays in microbial adaption and associated biogeochemical function, including the response to changes in timing and magnitude of flow variations.

The need for models that can incorporate the microbial response to changes in flow variation is particularly important in the
context of climate-watershed interactions. In years with low snowpack, the degree of hydropeaking through the summer is significantly elevated (Fig. 5). Local inundation dynamics (e.g., frequency of inundation or "return–interval") are therefore sensitive to watershed hydrology even in highly managed main-stem river corridor systems, and the influences should be even stronger in lower-order and unmanaged streams. Therefore, in watersheds that are projected to shift from snowpack to rain dominated hydrology, like those found in the Pacific Northwest, prediction of biogeochemical responses to changes in
inundation dynamics is needed. Filling this knowledge gap is urgent, given the global expansion of hydropower, with 3700 large dams planned worldwide as of 2015 (Zarfl et al., 2015), but also because of its widespread applicability to naturally variable systems (e.g., intermittent streams, unregulated rivers, tidally-influenced littoral zones of the Great Lakes).

## 5 Conclusion

From the synthesis of our results, we have derived the following conceptual model (Fig. 6): Watershed-climate interactions
lead to changes in hydropeaking dynamics, which alter parafluvial inundation dynamics (e.g., magnitude and time between inundation events). Irregular hydrologic fluctuations create spatial variation in the concentrations of C and N reactants and microbial communities adapted to local conditions (i.e., increasing fungal populations with increasing elevation and return–interval). Under conditions that allow for local adaptation, there is a relatively high flux of $CO_2$ from microbial respiration (left-hand side of Fig. 6). Upon re-wetting (right-hand side of Fig. 6), substrates become more available for use, but
microbial communities are initially suppressed metabolically because they are adapted to non-inundated conditions (e.g., rapid transport and significant penetration of oxygen into sediment due to gas filled pore spaces). The result is lower $CO_2$ flux rates primarily due, we hypothesize, to suppression of fungal respiration. Following prolonged inundation, the microbial community should adapt via a shift in composition away from fungal dominance. The time scales of such responses are an important knowledge gap and may depend upon the time-since-inundation. Ultimately, local adaptation to an inundated state
should lead to recovery of pre-inundation $CO_2$ flux rates, but the timescale of adaptation is unclear. This conceptual model hypothesizes that biogeochemical function (i.e., $CO_2$ flux rates) are resilient in the sense that they recover to pre-disturbance conditions. The degree to which this occurs, the associated time scales, and the dependence of 'biogeochemical resilience' on historical and ongoing inundation dynamics are important, open questions.

The Columbia River is one of many high order rivers with an extensive alluvial hyporheic zone (Downing et al., 2012). Our results are therefore applicable not only to other regions along the Columbia River, but also to other rivers worldwide. We demonstrate that carbon cycling in the parafluvial zone does not follow behavior commonly observed in soils and sediments,

wherein rewetting produces a pulse of $CO_2$, and that inundation history influences the ability of parafluvial hyporheic microbial communities to respond to re-inundation. In addition to causing spatial variation in the concentrations of key reactants within C and N cycles, inundation history along the parafluvial zone affects aquatic-terrestrial carbon cycling by driving variation in microbial community composition that, in turn, governs biogeochemical responses to hydrologic perturbation. The parafluvial hyporheic zone should therefore be considered as an important ecotone for biogeochemical dynamics and may need to be integrated as a distinct environment within hydrobiogeochemical models to predict the watershed biogeochemical function.

**Author contribution**

A. E. Goldman, E. B. Graham, A. R. Crump, D. W. Kennedy, and J. C. Stegen contributed to experimental design. A. E. Goldman, E. B. Graham, A. R. Crump, D. W. Kennedy, E. B. Romero, C. G. Anderson, K. L. Dana, C. T. Resch, J. K. Fredrickson contributed to sample collection and analysis. A. E. Goldman, E. B. Graham, A. R. Crump, D. W. Kennedy, J. K. Fredrickson, and J. C. Stegen contributed to data interpretation. A. E. Goldman prepared manuscript with contributions from all authors. The authors declare that they have no conflict of interest.

**Acknowledgments**

Thank you to Mike Perkins for Fig. 6 graphic, Hyun-Seob Song for modeling expertise, Eric Bottos, Taniya Roy Chowdhury, Yukari Maezato, and Tom Wietsma for sample analysis. This research was supported by the US Department of Energy (DOE), Office of Biological and Environmental Research (BER), as part of Subsurface Biogeochemical Research Program's Scientific Focus Area (SFA) at the Pacific Northwest National Laboratory (PNNL). PNNL is operated for DOE by Battelle Memorial Institute under contract DE–AC06–76RLO 1830.

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





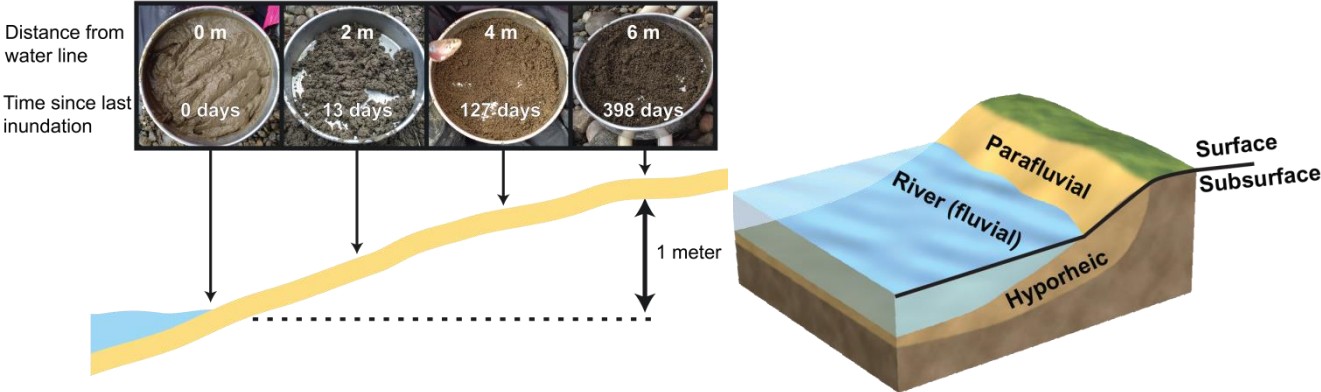

5    **Figure 1: Diagram of parafluvial hyporheic zone and of sampling design with photographs of sieved sediment sampled. The four locations were spaced laterally at 2-m intervals beginning at the water line and encompassed 1 m of elevation change. Note changes in sediment moisture and texture with changes in elevation. Color differences are due to camera quality and light availability and do not reflect sediment characteristics.**

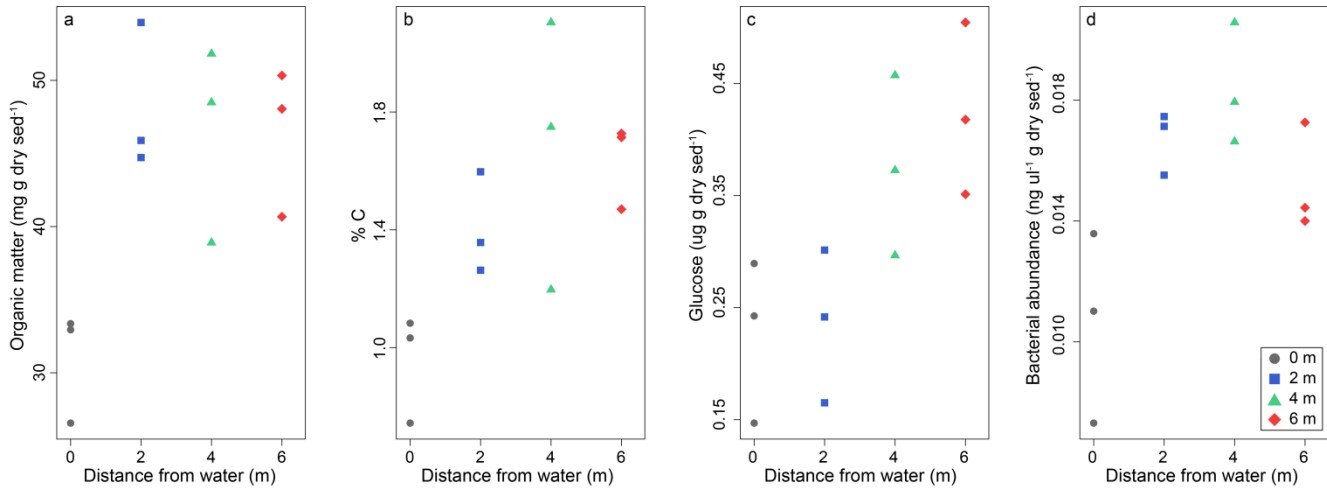

**Figure 2: Environmental/microbial variables with a stepped relationship to distance from water line. All have significant categorical differences (ANOVA; P<0.05 for all).**





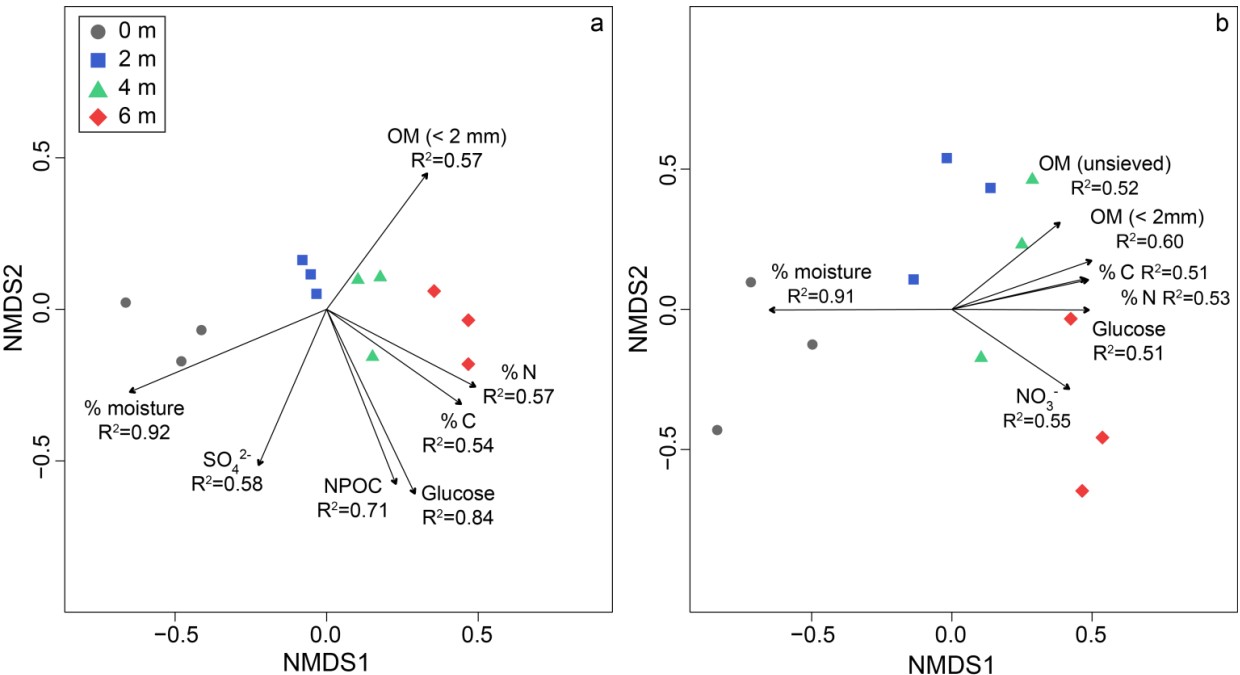

**Figure 3: Non-metric multidimensional scaling (NMDS) plots of bacteria/archaea (a) and fungi (b) community composition (n=12). Colors indicate groupings by elevations, which are significantly different (PERMANOVA, bacteria/archaea: P<0.001; fungi: P<0.01). Vectors indicate significant environmental variables with corresponding $R^2$ values.**

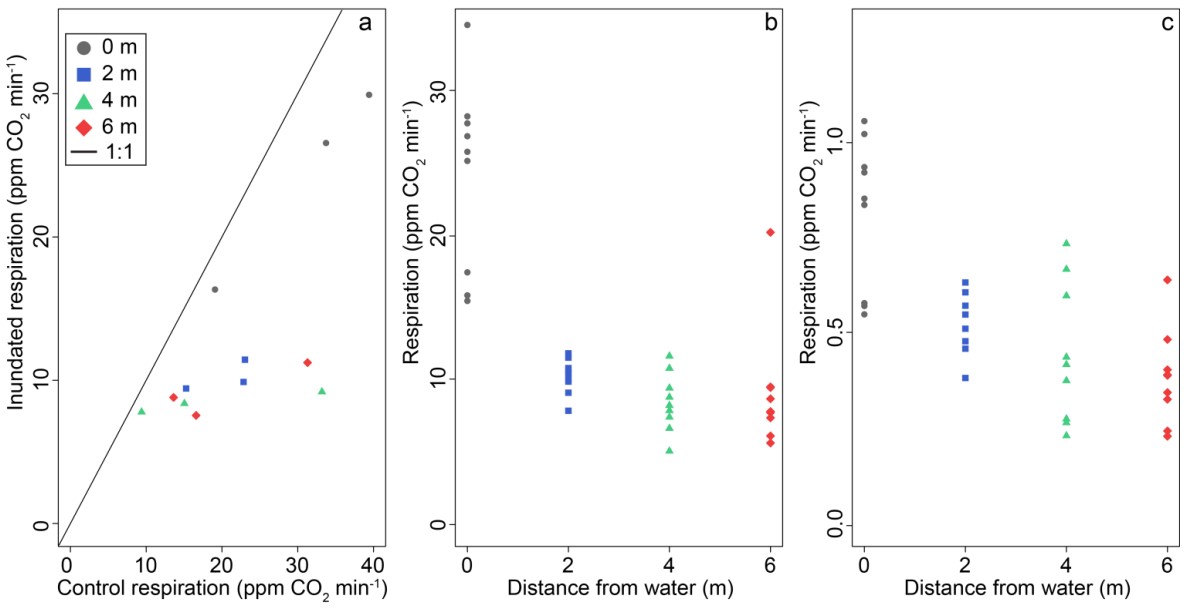

**Figure 4: RR of untreated, control samples and 0.5 hour inundated samples with black 1:1 line (a), 0.5 hour inundated RR and distance from water line (b), and 25 hour RR and distance from water line (c). Note that in the first plot, 0 m elevation RR (gray) plots along 1:1 line, but upper elevation inundated RR does not co-vary with uninundated RR.**





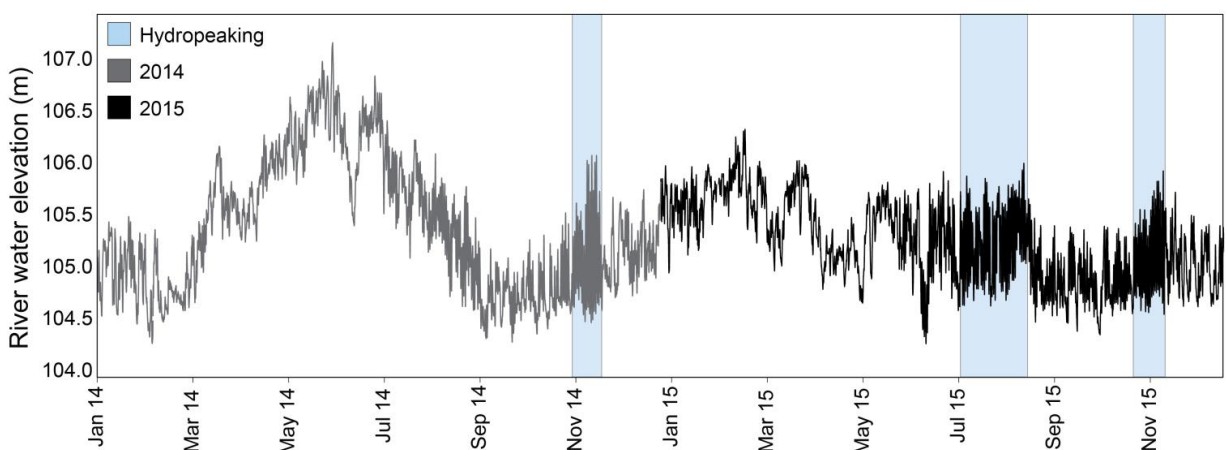

**Figure 5: River water elevation near the study site 2014 through 2015. Note the increase in water level elevation during the spring of 2014, caused by melting snowpack and the subsequent spring freshet. In 2015, the lack of spring freshet was due to very little snowpack in the winter, which then led to increased hydropeaking in the summer of 2015 that is typically only seen in early winter.**

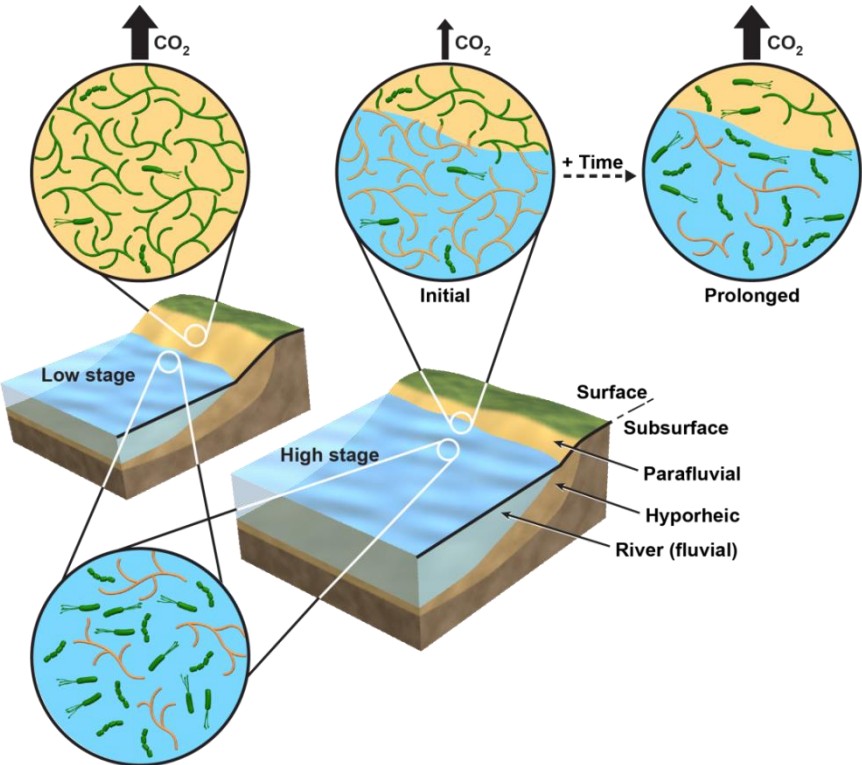

**Figure 6: Fungi (branching organisms) perform well (green) and increase in non-inundated conditions, but are suppressed (orange) upon re-inundation (blue shading), leading to lower $CO_2$ flux (thin arrow). We propose that prolonged inundation shifts the community towards bacterial (rod-shapes) dominance. The influence of inundation history on the timescale of the transient, metabolically suppressed, state is unknown.**





**Table 1. Mean, standard deviation (sd), significance (P), and $R^2$ values (n.s. if not significant) from linear regressions of environmental variables and distance from water line from untreated sediment (n=12). Organic matter (<2mm) also had a significant quadratic fit (P<0.01 (x); P=0.03 ($X^2$); $R^2$=0.57), which is not included below. All units are per gram dry sediment.**

|  | Mean | SD | P | $R^2$ |
|---|---|---|---|---|
| % moisture | 24.73 | 7.49 | <0.01 | 0.77 |
| Organic matter (<2 mm) (mg) | 42.97 | 8.56 | 0.04 | 0.30 |
| Organic matter (unsieved) (mg) | 32.82 | 6.38 | 0.17 | n.s. |
| Glucose (μmol) | 0.32 | 0.11 | 0.02 | 0.36 |
| Acetate (μmol) | 0.88 | 1.12 | 0.27 | n.s. |
| $NO_3^-$ (mg) | 0.082 | 0.122 | <0.01 | 0.47 |
| $NH_4^+$ (mg) | 0.010 | 0.0030 | 0.89 | n.s. |
| % C | 1.42 | 0.38 | <0.01 | 0.46 |
| % N | 0.14 | 0.038 | <0.01 | 0.47 |
| C/N | 9.89 | 0.47 | 0.94 | n.s. |
| Fungal to bacterial ratio | 0.058 | 0.041 | <0.01 | 0.50 |
| Fungal biomass (ng μL$^{-1}$) | 0.015 | 0.0035 | 0.01 | 0.45 |
| Bacterial biomass (ng μL$^{-1}$) | $9.4 \times 10^{-4}$ | $7.70 \times 10^{-4}$ | 0.09 | n.s. |
| Archaeal biomass (ng μL$^{-1}$) | $1.1 \times 10^{-4}$ | $6.9 \times 10^{-5}$ | 0.9 | n.s. |

**Table 2. Nitrate concentrations from untreated sediment (mean and standard deviation; uncorrected for grams of dry sediment) and treatment water.**

|  | $NO_3^-$ (mg/L) | SD |
|---|---|---|
| River water treatment | 1.75 | |
| Groundwater without $NO_3^-$ treatment | 0.70 | |
| Groundwater with $NO_3^-$ treatment | 14.49 | |
| 0 m sediment | 1.17 | 0.69 |
| 2 m sediment | 3.43 | 3.67 |
| 4 m sediment | 18.49 | 6.61 |
| 6 m sediment | 64.81 | 48.30 |