# Peer review of "Biogeochemical cycling at the aquatic-terrestrial interface is linked to parafluvial hyporheic zone inundation history"

_Biogeosciences, 2017_

## Referee Comment (RC1) · Anonymous Referee #1 · 13 May 2017

Review comments on bg-2017-28-manuscript-version2.pdf Dear Editor, While I found this paper of interest, generally well written with data well presented, I have several main concerns. First of all, the authors should use such unusual setting (anthropogenic driven soil saturation which occur on very limited areas of emerged Earth to allow the better understanding of the relationship between soil water content and soil saturation by water and major biogeochemical cycles. This is not performed and when done, it should greatly enhance the impact of the performed research. In their acknowledgment of the existing literature the authors should present quantitative information, trends from studies having investigated the impact of soil inundation of the cycles of elements and discuss the trends for further identification of gaps. While for instance water satura-

tion by water bodies leads in many cases to denitrification in sediments, Grimaldi and Chaplot (Water, Air, and Soil Pollution, 2000) showed that in some cases exchange processes between streamwater, the hyporheic and riparian zones forbid that process. The second issue concerns the interpretation of the results. Not only are the study variables of interest (microbial biomass,. . .) affected by inundation but also by multiple factors such as inerrant soil type, vegetation type and its impact on rooting, organic matter quality, soil temperature,. . ..This is a major which by itself nullifies all conclusions. Moreover, how can the authors convince on the observed trends is no temporal evaluation has been performed? The third issue concerns the study objectives. The authors should decide about their objectives. Is it about process understanding or about modeling?. Authors have to choice. I suggest to move all the conceptual results to the discussion part of the paper. Figure 2 is too simplistic. Other comments Figure 1 can not be presented with not true colors or pictures taken under constant light. In figure 2 why having a legend with distance from water? Why colors into the figure? This "distance to water" is not really adequate as the water level always changes.

―――――――――――――――――――――

---

## Referee Comment (RC3) · Anonymous Referee #2 · 25 May 2017

I find the article generally interesting. It is well written, includes extensive literature documentation, and pays ample attention to describe the practical and general implications of this study which is undertaken in a rather unusual, dam-controlled river environment.

Coming from the geochemistry side of biogeochemistry, my comments focus on the physico/chemical part of the study. There are two topics I have problems with. The first is temperature, or rather the lack of its consideration (only "room temperature" is mentioned twice on page 4). Are the results temperature-independent? If so, the authors should explain it. If the results are temperature dependent, the authors will have to discuss the implications of it, and preferably give some more accurate temperature conditions during the experiments.

The second problem I have with the paper is that on page 5, lines 5-7, the authors describe in detail how measurements of cations and anions are conducted. However, none of those parameters, with the exception of nitrate or nitrite, is ever mentioned in the paper – so, why are the measurements listed? If the chemistry plays no role, it should be mentioned; if the authors discuss it in a companion paper, it should be mentioned; if the data were not used at all, their measurement methods should not be discussed in the methods section.

As far as figures are concerned, I would suggest two changes. In Figure 1 the colors are described to be misleading, so why show them? The photos should be converted to grey scale. Figure 6 could be deleted, it is very simplistic and the importance of changing biology is extensively described, with much more detail, in the text of the publication.

---

## Author Comment (AC2) · 23 Jun 2017

Please find in the attached supplement a detailed response to each reviewer.

———————————————

---

## Author Comment (AC3) · 23 Jun 2017

Please find in the attached supplement a detailed response to each reviewer.

---

## Author Response (AR2)

Revised Manuscript Submission

bg-2017-28

Associate Editor Review and Response

August 2017

Associate Editor Comments

**Associate Editor Decision: Publish subject to minor revisions (Editor review)** (01 Aug 2017)
by Clare Woulds
Comments to the Author:
Dear Dr Goldman

Thank you for your revised manuscript. I am happy with the way that you have addressed most of the reviewer comments. I would however like to request further changes to figure 6, as I feel it is slightly inconsistent. Firstly, the two hyporheic zone block drawings should be the same size, and side by side (rather than one set back, and therefore smaller). Secondly, I feel that an arrow with width indicating $CO_2$ emissions should be added to the circle representing submerged sediment in both scenarios. To be fully clear, I'd actually suggest having two versions of the circle for submerged sediment, one for each block. Perhaps most importantly, I'm not really sure how the figure represents the different response to re-wetting in recently and less recently submerged sediment. Should you actually have four blocks in the diagram, rather than just two? Is this what the '+ time' dotted line is supposed to represent?

If you have any questions please let me know.

Regards,

Clare Woulds

Author Response to Associate Editor Comments

bg-2017-28

Please find in plain text our response to the associate editor's review. The editor's comments are in bold.

**I would however like to request further changes to figure 6, as I feel it is slightly inconsistent. Firstly, the two hyporheic zone block drawings should be the same size, and side by side (rather than one set back, and therefore smaller). Secondly, I feel that an arrow with width indicating CO2 emissions should be added to the circle representing submerged sediment in both scenarios. To be fully clear, I'd actually suggest having two versions of the circle for submerged sediment, one for each block. Perhaps most importantly, I'm not really sure how the figure represents the different response to re-wetting in recently and less recently submerged sediment. Should you actually have four blocks in the diagram, rather than just two? Is this what the '+ time' dotted line is supposed to represent?**

We agree that Figure 6 can be improved upon. With this in mind, we have made a number of changes to Figure 6 that we believe address your concerns about the clarity and consistency across the figure components. Below is the revised figure and associated caption. We have added a full set of panels for both the "not inundated at time of sampling" sediment (middle) and the "inundated at time of sampling" sediment (far right) and have included $CO_2$ arrows on all of the panels that represent our experimental incubations (b,c,d). We have also removed the "+ time" notation and have instead separated out the initial and prolonged response to inundation. Accordingly, we have made small changes to the manuscript text that references Fig. 6 so that the information corresponds with the appropriate components of the figure.

[revised manuscript text omitted]

**Not inundated at time of sampling**

**Inundated at time of sampling**

**a** Sampling conditions
Surface
Subsurface
Parafluvial
Hyporheic
River (fluvial)

**b** Starting experimental conditions
$CO_2$ · $CO_2$

**c** Initial inundation conditions
$CO_2$ · $CO_2$

**d** Prolonged inundation conditions
$CO_2$ · $CO_2$

Low stage · High

**Figure 6:** Illustration of field conditions during sampling (a) and of incubation conditions prior to experimental inundation (b), during initial inundation (c), and following prolonged inundation (d). River elevation diagrams (far left) display field conditions replicated by the incubation experiments. In historically uninundated sediments (middle), fungi (branching organisms) perform well (green) (a,b), but are metabolically suppressed (orange) upon initial re-inundation (blue shading), leading to lower CO2 flux (thin arrow) (c). We propose that prolonged inundation shifts the community towards bacterial (rod-shapes) dominance, leading to an eventual recovery of CO2 flux (thick arrow) (d). In historically inundated sediments (far right), bacteria perform well and maintain a higher CO2 flux both during a brief drop in river stage in which the sediments lack an overlying water column but stay saturated (b) and when water level rises (c,d).

**Table 1. Mean, standard deviation (sd), significance (P), and $R^2$ values (n.s. if not significant) from linear regressions of environmental variables and the log of days since inundation from untreated sediment (n=12). Organic matter (<2mm) also had a significant quadratic fit (P<0.01 (x); P=0.03 ($X^2$); $R^2$=0.57), which is not included below. All units are per gram dry sediment.**

| | Mean | SD | P | $R^2$ |
|---|---|---|---|---|
| % moisture | 24.73 | 7.49 | <0.01 | 0.77 |
| Organic matter (<2 mm) (mg) | 42.97 | 8.56 | 0.02 | 0.41 |
| Organic matter (unsieved) (mg) | 32.82 | 6.38 | 0.12 | n.s. |
| Glucose (μmol) | 0.32 | 0.11 | <0.01 | 0.53 |
| Acetate (μmol) | 0.88 | 1.12 | 0.22 | n.s. |
| $NO_3^-$ (mg) | 0.082 | 0.122 | 0.02 | 0.38 |
| $NH_4^+$ (mg) | 0.010 | 0.0030 | 0.93 | n.s. |
| % C | 1.42 | 0.38 | <0.01 | 0.54 |
| % N | 0.14 | 0.038 | <0.01 | 0.55 |
| C/N | 9.89 | 0.47 | 0.87 | n.s. |
| Fungal to bacterial ratio | 0.058 | 0.041 | 0.01 | 0.43 |
| Fungal biomass (ng μL$^{-1}$) | 0.015 | 0.0035 | 0.01 | 0.42 |
| Bacterial biomass (ng μL$^{-1}$) | $9.4 \times 10^{-4}$ | $7.70 \times 10^{-4}$ | 0.03 | 0.32 |
| Archaeal biomass (ng μL$^{-1}$) | $1.1 \times 10^{-4}$ | $6.9 \times 10^{-5}$ | 0.85 | n.s. |

**Table 2. Nitrate concentrations from untreated sediment (mean and standard deviation; uncorrected for grams of dry sediment) and nitrate and anion concentrations from treatment water.**

| | $NO_3^-$ (mg/L) | $Na^+$ (mg/L) | $Ca^{2+}$ (mg/L) | $Mg^{2+}$ (mg/L) | $K^+$ (mg/L) |
|---|---|---|---|---|---|
| River water treatment | 1.75 | 4.16 | 19.85 | 5.07 | 1.05 |
| Groundwater without $NO_3^-$ treatment | 0.70 | 18.35 | 29.80 | 8.70 | 3.83 |
| Groundwater with $NO_3^-$ treatment | 14.49 | 18.14 | 30.91 | 8.54 | 4.28 |

| | $NO_3^-$ (mg/L) | SD |
|---|---|---|
| 0 d sediment | 1.17 | 0.69 |
| 13 d sediment | 3.43 | 3.67 |
| 127 d sediment | 18.49 | 6.61 |
| 398 d sediment | 64.81 | 48.30 |

Peer Reviews and Response

July 2017

Reviewer Comments

Biogeosciences Discuss.,
doi:10.5194/bg-2017-28-RC1, 2017

[Figure]
Review comments on bg-2017-28-manuscript-version2.pdf Dear Editor, While I found this paper of interest, generally well written with data well presented, I have several main concerns. First of all, the authors should use such unusual setting (anthropogenic driven soil saturation which occur on very limited areas of emerged Earth to allow the better understanding of the relationship between soil water content and soil saturation by water and major biogeochemical cycles. This is not performed and when done, it should greatly enhance the impact of the performed research. In their acknowledgment of the existing literature the authors should present quantitative information, trends from studies having investigated the impact of soil inundation of the cycles of elements and discuss the trends for further identification of gaps. While for instance water satura-

tion by water bodies leads in many cases to denitrification in sediments, Grimaldi and Chaplot (Water, Air, and Soil Pollution, 2000) showed that in some cases exchange processes between streamwater, the hyporheic and riparian zones forbid that process. The second issue concerns the interpretation of the results. Not only are the study variables of interest (microbial biomass,. . .) affected by inundation but also by multiple factors such as inerrant soil type, vegetation type and its impact on rooting, organic matter quality, soil temperature,. . ..This is a major which by itself nullifies all conclusions. Moreover, how can the authors convince on the observed trends is no temporal evaluation has been performed? The third issue concerns the study objectives. The authors should decide about their objectives. Is it about process understanding or about modeling?. Authors have to choice. I suggest to move all the conceptual results to the discussion part of the paper. Figure 2 is too simplistic. Other comments Figure 1 can not be presented with not true colors or pictures taken under constant light. In figure 2 why having a legend with distance from water? Why colors into the figure? This "distance to water" is not really adequate as the water level always changes.

—————————————————————

[Figure]

Biogeosciences Discuss.,
doi:10.5194/bg-2017-28-RC3, 2017

[Figure]

I find the article generally interesting. It is well written, includes extensive literature documentation, and pays ample attention to describe the practical and general implications of this study which is undertaken in a rather unusual, dam-controlled river environment.

Coming from the geochemistry side of biogeochemistry, my comments focus on the physico/chemical part of the study. There are two topics I have problems with. The first is temperature, or rather the lack of its consideration (only "room temperature" is mentioned twice on page 4). Are the results temperature-independent? If so, the authors should explain it. If the results are temperature dependent, the authors will have to discuss the implications of it, and preferably give some more accurate temperature conditions during the experiments.

[Figure]

The second problem I have with the paper is that on page 5, lines 5-7, the authors describe in detail how measurements of cations and anions are conducted. However, none of those parameters, with the exception of nitrate or nitrite, is ever mentioned in the paper – so, why are the measurements listed? If the chemistry plays no role, it should be mentioned; if the authors discuss it in a companion paper, it should be mentioned; if the data were not used at all, their measurement methods should not be discussed in the methods section.

As far as figures are concerned, I would suggest two changes. In Figure 1 the colors are described to be misleading, so why show them? The photos should be converted to grey scale. Figure 6 could be deleted, it is very simplistic and the importance of changing biology is extensively described, with much more detail, in the text of the publication.

———————————————————

[Figure]

Author Response to Comments

Biogeosciences Discuss.,
https://doi.org/10.5194/bg-2017-28-AC1, 2017

[Figure]

Author response to reviewer comments on "Carbon cycling at the aquatic-terrestrial interface is linked to parafluvial hyporheic zone inundation history" by A. E. Goldman et al.

Please find in plain text our responses to both reviews. Reviewer comments are in bold.

**RC1 and RC2 (duplicate comment; 13 and 14 May 2017):**
**Dear Editor, While I found this paper of interest, generally well written with data well presented, I have several main concerns.**

**First of all, the authors should use such unusual setting (anthropogenic driven soil saturation which occur on very limited areas of emerged Earth to allow the better understanding of the relationship between soil water content and soil saturation by water and major biogeochemical cycles. This is not performed and when done, it should greatly enhance the impact of the performed research.**
Thank you for your comment. While we agree that further investigating the nuanced relationship between varying soil water content and biogeochemical cycles is merit worthy, this research focuses on the historical influence of river inundation on sediment. Although water saturation is an important driver of the environmental changes we investigate, a detailed analysis of different sediment water contents is outside the scope of this research.

We also note that while anthropogenically driven sediment saturation may initially seem to be an unusual setting, dams are widespread and globally distributed. In the United States, 2653 dams generate hydropower (USACE 2016) and 90% of water discharge is hydrologically altered (Jackson et al., 2001). Hydropower currently accounts for 80% of renewable energy globally (Zarfl et al., 2015; Hermoso, 2017) and is projected to increase as countries begin managing their energy production within the Paris Agreement and other climate accords (Hermoso, 2017; Latrubesse et al., 2017).

**In their acknowledgment of the existing literature the authors should present quantitative information, trends from studies having investigated the impact of soil inundation of the cycles of elements and discuss the trends for further identification of gaps. While for instance water saturation by water bodies leads in many cases to denitrification in sediments, Grimaldi and Chaplot (Water, Air, and Soil Pollution, 2000) showed that in some cases exchange processes between streamwater, the hyporheic and riparian zones forbid that process.**

We agree that it is important to include quantitative results of literature pertinent to the scope of the research. Because the research we present is unique with regard to looking at the historical effects of inundation in the parafluvial zone (page 3, line 5-8), we are unable to put directly comparable results into our literature review. Instead, we utilize the discussion section to link specific qualitative and quantitative results from reviewed literature to specific results that we

present (page 8-11). By using this structure, we aim to minimize reader confusion and to use references in the most impactful way possible.

In order to keep the text up to date, we will add text to the discussion (Section 4.1 Microbial response to inundation), related to a May 2017 publication that indicates soil respiration is determined in part by the historical influence of climate, with historically wetter soil emitting twice as much carbon, on average, than historically drier soils (Hawkes et al., 2017). This conclusion aligns with our result of historically wetter sediment emitting more $CO_2$ that historically drier sediment.

**The second issue concerns the interpretation of the results. Not only are the study variables of interest (microbial biomass,…) affected by inundation but also by multiple factors such as inerrant soil type, vegetation type and its impact on rooting, organic matter quality, soil temperature,….This is a major which by itself nullifies all conclusions.**

We believe the reviewer is stating that the conclusions we present are weakened due to the many variables influencing the study system. We agree that many factors influence carbon cycling, nitrogen cycling, and microbial communities, but we note that this does not preclude interpretation of the system's response to inundation. The goal of this research was to investigate the influence of hydrologic history on the system response as a whole, which is described on page 2, line 23-24 ("By identifying the processes impacted by the historical contingencies of this variable inundation, it is possible to understand linkages among hydrology, biogeochemistry, and microbial ecology on a systems level"). In our discussion of the microbial respiration response (page 8, line 23-24), we also specifically mention that having a consistent respiration rate (RR) across the upper elevations was unexpected due to the environmental and microbial differences across the upper elevations ("Consistent RR across the upper elevations was unexpected, because they differed significantly in features that influence RR, including microbial composition, surrounding vegetation, and time since most recent river inundation"), and we subsequently present a hypothesis that aligns with our conceptual model ("We suggest that microbial communities in sediments that are inundated permanently or for extended periods are better adapted to full water saturation than those that are infrequently inundated, which led to the binary RR response between 0 m and the upper elevations.")

Over time, hydrologic history drives changes in vegetation, microbial community composition, physical sediment characteristics, and many other variables (Hupp and Osterkamp, 1996; Tockner et al., 2000; Larned et al., 2010). We describe this on page 2 (line 20-23): "Over time, cycles of wetting and drying impact different elevations of the parafluvial zone at different frequencies, which naturally results in a gradient of sediment moisture content, but also has the potential to create biogeochemical and microbial interactions specifically dependent on preceding environmental conditions." By studying the whole system response, we do not attempt to resolve the intermediate mechanisms. Instead, we propose a conceptual model and hypotheses that can help guide future research to resolve the mechanisms at play. Rather than weakening our results, the integration of variables influencing the study system helps us to investigate the complexity of the natural system and creates a foundation for future research. We appreciate the reviewer's comment, and we will add the references listed above to the introduction to indicate

the context for our statements regarding hydrologic history's influence on many environment and microbial varabiables.

**Moreover, how can the authors convince on the observed trends is no temporal evaluation has been performed?**

Thank you for highlighting that we do not adequately discuss the importance of temporal influence in our text. We recognize that our study utilizes samples from a single point in time, a common limitation in scientific studies. We will specify in our conclusions that seasonal differences are an additional question to investigate and that future work would benefit from studying other dam-influenced systems to investigate if other systems behave as our study site does.

**The third issue concerns the study objectives. The authors should decide about their objectives. Is it about process understanding or about modeling?. Authors have to choice. I suggest to move all the conceptual results to the discussion part of the paper.**

Thank you for your comment. The study objectives are listed in Section 1 (page 3, line 11-15), and accompanying hypotheses are listed with the objectives (page 3, line 18-26). Our objectives are focused on process understanding. We mention modeling at two points in the paper. In the introduction, we briefly mention modeling to identify the context and implications of this work (page 2, line 25). Additionally, Section 4.3 is designated for discussing linkages between this work and modeling (page 10, lines 17-31; page 11 line 1-12), because modeling is an important component of the implications and transferability of this interdisciplinary study. Our conclusions (Section 5, page 11, line 13-32; page 12, line 1-7), however, focus on process understanding via the conceptual model that we outline.

In Section 4.3, we aim to establish context for the current state of predictive models, explain how our results identify modeling research needs, and illustrate the current importance of moving modeling research forward. We will make several changes to more clearly present our aims within the text. We will change the section heading from "Importance of parafluvial hyporheic dynamics to hydrobiogeochemical models," to "Implications of parafluvial hydrobiogeochemical processes for predictive models." We will also revise the text (page 10, line 18-31; page 11, line 1-2) to better integrate our results with the modelling contex as indicated below (changes in bold and strikethrough):

"Biogeochemical response lags due to physiological stress of parafluvial microbial communities are of particular importance in creating modeling frameworks that can accurately predict the biogeochemical behavior of terrestrial-aquatic transition zones in managed systems. **Our results indicate that a combination of ecological and physiological mechanisms limited the ability of the microbial communities to rapidly adapt to an inundated state. This resulted in the suppression of RR, which we propose is a transient condition whereby there is also temporal lag in biogeochemical function.**

environment is not instantaneous in general, but a **A** long time lag is often observed **in microbial adaptation to the abrupt change in local environment** (Wood et al., 1995). This phenomenon can affect the transport of biodegradable substrates and contaminants, particularly when the timescale of the metabolic lag is comparable to that of transport (Nilsen et al., 2012). Despite such importance, mechanistic approaches for incorporating metabolic lags into subsurface transport modeling are currently lacking. Past modeling work that attempted to account for this lag effect was either based on temporal convolution integral (Wood et al., 1995; Nilsen et al., 2012) or introducing exposure time as an additional dimension (or coordinate) (Ginn, 1999). These efforts do not address fundamental mechanisms responsible for time lags and consequently, their predictions may be limited.  **Given that we find time lags to be extremely important to the microbial and biogeochemical behavior of our system,** future **modeling** efforts need to consider more ecologically and physiologically relevant causes for delays in microbial adaption and associated biogeochemical function, including the response to changes in timing and magnitude of flow variations."

**Other comments Figure 1 can not be presented with not true colors or pictures taken under constant light.**

We agree. To address concerns of both reviewers, we will change the figure so that the photographs are presented in greyscale to minimize the differences in lighting/camera quality. We feel it is important to keep the photographs so that readers can visually see differences in sediment moisture and texture across the sampling locations.

[Figure]

**Figure 2 is too simplistic. In figure 2 why having a legend with distance from water? Why colors into the figure? This "distance to water" is not really adequate as the water level always changes.**

Thank you for your comment. We chose to use both different shapes and different colors to differentiate the sampling locations. We kept the color scheme consistent throughout all figures, which allows the colors to aid in visualization across the different figures. Using greyscale colors would make it more difficult to make visual linkages across the different figures presented in the manuscript, and we use diferent shapes to assist colorblind readers.

"Distance to water" refers to the distance to the water line at the time of sampling and is therefore a stable value. In order to more clearly differentiate among sample locations and to change the focus to the primary research question- the historical influence of inundation- we will change the terminology throughout the text and figures to utilize days since last inundation (0, 13, 127, 398 days) instead of distance to water (0, 2, 4, 6 m). For brevity, we will still include references to "elevation" as a means of differentiating among sample sites. Making this change will impact Table 1, which presents results from linear regressions. Although the exact p and $R^2$ values change in the new regressions, using days since last inundation does not alter whether or not the variable has a significant relationship to the sampling location except in one instance. Bacterial biomass does not have a significant relationship to distance from water (p=0.09; $R^2$=0.19), but it does have a significant relationship to days since inundation (p=0.03; $R^2$=0.32). Subsequently, we will change the text that mentions the result explicitly (page 7, line 4-8). Making this change will not influence any of our conclusions.

**RC3 (25 May 2017):**

**I find the article generally interesting. It is well written, includes extensive literature documentation, and pays ample attention to describe the practical and general implications of this study which is undertaken in a rather unusual, dam-controlled river environment. Coming from the geochemistry side of biogeochemistry, my comments focus on the physico/chemical part of the study. There are two topics I have problems with.**

**The first is temperature, or rather the lack of its consideration (only "room temperature" is mentioned twice on page 4). Are the results temperature-independent? If so, the authors should explain it. If the results are temperature dependent, the authors will have to discuss the implications of it, and preferably give some more accurate temperature conditions during the experiments.**

We agree that temperature is an important component that we did not discuss adequately. We will add text to the methods (Section 2.1 Study Site) that characterizes the temperatures of the study site. We will include historical (2014-2016) river water temperature, mean groundwater temperature, and air temperature. We will also include the river water temperature at the start of sample collection and the air temperature from the date of sample collection. Additionally, we will remove the term "room temperature" (page 4, line 23 and 31) and instead state that sample processing and incubations were performed in a laboratory maintained at temperatures between 21 and 22°C, which correspond to maximum river temperatures. We will also clarify in the methods (Section 2.3 Laboratory incubations) that the river water used in incubations was left at ambient laboratory temperature for approximately one hour prior to use.

As the reviewer alludes to, process rates are likely temperature dependent. With this in mind, we discuss our results in a relative, as opposed to absolute, sense throughout the text. We assume that temperature had the same effect across samples, which is supported by the randomized sample processing order, which we will add text to the methods to explain (Section 2.3 Laboratory incubations). We note, however, that the lack of temperature control precludes the comparison of our data to other work in an absolute, rather than relative, sense, and we will add text to the methods that explains this important point.

**The second problem I have with the paper is that on page 5, lines 5-7, the authors describe in detail how measurements of cations and anions are conducted. However, none of those parameters, with the exception of nitrate or nitrite, is ever mentioned in the paper – so, why are the measurements listed? If the chemistry plays no role, it should be mentioned; if the authors discuss it in a companion paper, it should be mentioned; if the data were not used at all, their measurement methods should not be discussed in the methods section.**

We agree that we can improve the clarity with which we discuss anions and cations. We will add additional text throughout the manuscript to more clearly present the anion results. Specifically, we will add text that states that $SO_4^{2-}$ was significant in the PERMANOVA results and that $SO_4^{2-}$ is displayed on the NMDS plots (Fig. 3)(Section 3.1.1 Microbial biogeography). We will also add text (Section 3.1.1 Spatial gradients in environmental variables) that states that there were no

significant relationships between the non-nitrogen anions (Cl$^-$, SO$_4^{2-}$) and time since last inundation.

Unfortunately, we do not have cation measurements from untreated, control samples, which we apologize for listing in error in the methods section (page 5, line 14) and which we will correct. We mention cations as a component of our motivations for utilizing three different treatment waters (page 3, line 15-26). We will add text (Section 3.2.1 Response to different treatment waters) that lists the differing cation content of the three treatment waters as an additional means of highlighting the lack of treatment effects (page 7, line 30), and we will list the associated values in Table 2. We will also clarify in the text ("The initial (0.5 hour), prolonged (25 hour), or change (25hr–0.5hr) in any measured inundation response did not differ significantly across treatment waters", page 7, line 28-29) which variables we are referring to, including cations. We will also add a supplemental table (Table S3) that provides the anion, cation, respiration rate, and NPOC data from the incubations, so that readers can easily access the information.

**As far as figures are concerned, I would suggest two changes. In Figure 1 the colors are described to be misleading, so why show them? The photos should be converted to grey scale.**

We agree with the assessment regarding the colors in Figure 1. To address concerns from both reviewers, we will change the photographs to grey scale to minimize the differences in lighting/camera quality. Please see page 12 for the new figure.

**Figure 6 could be deleted, it is very simplistic and the importance of changing biology is extensively described, with much more detail, in the text of the publication.**

We agree that Figure 6 simplistic. This was an intentional choice to allow easy visualization of the complex concepts described in the text. We believe it is an important addition to the manuscript, as it both summarizes our conceptual model and aids readers less familiar with the concepts discussed. In particular, the figure clearly illustrates the differences between the initial versus prolonged effect of inundation with regard to both microbial community and CO$_2$ flux. It also allows for the juxtaposition of the constantly-inundated versus variably-inundated sites and thus stresses the open question of timescales of biogeochemical resilience, which underpins the need for future research. If the figure was removed, it would be difficult for readers to see the whole conceptual framework that the study points towards. Losing that connection will diminish the scientific impact of the paper. Additionally, many readers flip through figures before reading a manuscript. With the conceptual figure, those readers can understand the major outcomes without needing to spend significant time reading the paper, which will make our results more widely accessible.

Revised Text

[revised manuscript text omitted]
 inundation history, and brown symbols indicate non-significant taxa. 0 d and 398 d elevations contribute 100% of taxa driving dissimilarity.

Table S3. Mean and standard deviation of NPOC, Cl⁻, SO₄²⁻, NO₃⁻, NH₄⁺, Na⁺, Ca²⁺, Mg²⁺, K⁺, and respiration rate from 0.5hr and 25hr incubations. Treatments are indicated by RW (river water), GW (groundwater without $NO_3^-$), and GWN (groundwater with $NO_3^-$). All units are per g dry sediment. N=3 for all treatments except 0 days/ 25hr/ river water (n=2).

| Time since inundated | Incubation | Treatment | NPOC (mg) Mean | SD | Cl (mg) Mean | SD | SO₄²⁻ (mg) Mean | SD | NO₃⁻ (mg) Mean | SD | NH₄⁺ (mg) Mean | SD | Na⁺ (mg) Mean | SD | Ca²⁺ (mg) Mean | SD | Mg²⁺ (mg) Mean | SD | K⁺ (mg) Mean | SD | Respiration rate (ppm CO₂/min) Mean | SD |
|---|---|---|---|---|---|---|---|---|---|---|---|---|---|---|---|---|---|---|---|---|---|---|
| 0 days | 0.5hr | RW | 5.85E-03 | 6.66E-04 | 7.96E-03 | 2.05E-03 | 2.28E-02 | 6.66E-03 | 2.21E-02 | 2.67E-02 | 1.63E-02 | 7.93E-03 | 8.51E-03 | 1.26E-03 | 6.76E-03 | 1.95E-03 | 1.52E-03 | 4.68E-04 | 2.02E-03 | 2.94E-04 | 23.06 | 6.31 |
| | 0.5hr | GW | 4.53E-03 | 9.06E-04 | 1.27E-02 | 5.38E-03 | 2.66E-02 | 7.32E-03 | 2.73E-02 | 4.00E-02 | 1.52E-02 | 4.82E-03 | 8.65E-03 | 2.57E-03 | 6.86E-03 | 3.01E-03 | 1.54E-03 | 6.07E-04 | 3.26E-03 | 1.02E-03 | 0.75 | 0.16 |
| | 0.5hr | GWN | 4.26E-03 | 1.17E-03 | 3.03E-02 | 1.97E-02 | 4.60E-02 | 2.55E-02 | 5.52E-02 | 8.68E-02 | 1.21E-02 | 4.34E-03 | 1.02E-02 | 3.53E-03 | 6.50E-03 | 2.43E-03 | 1.45E-03 | 4.87E-04 | 2.40E-03 | 1.64E-03 | 26.92 | 8.73 |
| | 25hr | RW | 5.04E-03 | 1.43E-03 | 3.23E-02 | 8.26E-03 | 4.19E-02 | 7.17E-03 | 5.86E-02 | 9.68E-02 | 1.63E-02 | 6.16E-03 | 1.19E-02 | 4.19E-03 | 8.65E-03 | 3.01E-03 | 1.95E-03 | 6.62E-04 | 4.02E-03 | 1.29E-03 | 0.88 | 0.27 |
| | 25hr | GW | 3.88E-03 | 1.07E-03 | 1.65E-02 | 5.38E-03 | 3.10E-02 | 7.93E-03 | 3.58E-02 | 4.69E-02 | 1.40E-02 | 5.11E-03 | 8.98E-03 | 3.23E-04 | 6.19E-03 | 4.92E-04 | 1.38E-03 | 5.24E-05 | 1.69E-03 | 2.25E-04 | 22.84 | 6.35 |
| | 25hr | GWN | 7.25E-03 | 2.26E-03 | 2.72E-02 | 8.98E-03 | 4.35E-02 | 8.25E-03 | 4.64E-02 | 7.04E-02 | 1.68E-02 | 4.31E-03 | 1.42E-02 | 2.07E-03 | 8.50E-03 | 2.74E-03 | 1.86E-03 | 7.53E-04 | 4.46E-03 | 1.72E-03 | 0.80 | 0.22 |
| 13 days | 0.5hr | RW | 5.40E-03 | 1.33E-03 | 5.60E-03 | 2.65E-03 | 1.48E-02 | 3.81E-03 | 2.12E-02 | 2.06E-02 | 1.61E-02 | 2.54E-03 | 8.42E-03 | 3.41E-03 | 1.19E-02 | 1.21E-02 | 1.18E-03 | 1.92E-04 | 1.95E-03 | 4.13E-04 | 9.53 | 1.88 |
| | 0.5hr | GW | 8.88E-03 | 2.96E-03 | 7.76E-03 | 6.11E-04 | 1.49E-02 | 1.20E-03 | 2.26E-02 | 2.83E-02 | 2.26E-02 | 6.49E-03 | 5.86E-03 | 2.16E-03 | 8.20E-03 | 1.34E-04 | 1.92E-03 | 8.62E-05 | 3.00E-03 | 4.79E-04 | 0.48 | 0.04 |
| | 0.5hr | GWN | 5.51E-03 | 1.83E-03 | 1.96E-02 | 1.11E-03 | 2.59E-02 | 1.80E-03 | 3.86E-02 | 5.28E-02 | 1.63E-02 | 3.21E-03 | 6.59E-03 | 9.84E-04 | 7.12E-03 | 3.56E-03 | 1.65E-03 | 8.91E-04 | 2.41E-03 | 1.14E-03 | 10.61 | 1.10 |
| | 25hr | RW | 7.95E-03 | 3.74E-03 | 2.30E-02 | 1.78E-03 | 3.00E-02 | 1.92E-03 | 4.43E-02 | 7.20E-02 | 2.06E-02 | 4.31E-03 | 6.32E-03 | 3.48E-04 | 1.10E-02 | 4.62E-03 | 1.96E-03 | 3.08E-04 | 2.67E-03 | 7.83E-04 | 10.61 | 0.09 |
| | 25hr | GW | 5.53E-03 | 1.70E-03 | 1.26E-02 | 9.23E-03 | 2.14E-02 | 1.57E-02 | 7.73E-03 | 5.58E-03 | 9.99E-03 | 7.27E-03 | 6.75E-03 | 6.82E-04 | 5.60E-03 | 2.71E-04 | 1.23E-03 | 4.95E-05 | 1.79E-03 | 6.08E-05 | 10.61 | 0.21 |
| | 25hr | GWN | 1.09E-02 | 5.84E-03 | 2.08E-02 | 4.69E-03 | 2.93E-02 | 1.79E-02 | 2.90E-03 | 1.39E-03 | 2.36E-02 | 4.75E-03 | 6.17E-03 | 5.46E-04 | 1.12E-02 | 5.02E-03 | 2.80E-03 | 1.27E-03 | 4.03E-03 | 1.01E-03 | 0.59 | 0.04 |
| 127 days | 0.5hr | RW | 9.49E-03 | 5.93E-03 | 7.50E-03 | 4.15E-04 | 2.04E-02 | 3.02E-03 | 5.55E-02 | 2.65E-02 | 1.88E-02 | 1.17E-02 | 6.81E-03 | 1.91E-03 | 8.90E-03 | 3.65E-03 | 2.17E-03 | 1.10E-03 | 1.02E-02 | 5.04E-03 | 8.57 | 2.12 |
| | 0.5hr | GW | 1.05E-02 | 6.46E-03 | 8.16E-03 | 7.47E-04 | 1.67E-02 | 4.95E-04 | 1.05E-02 | 3.20E-03 | 2.47E-02 | 6.09E-03 | 7.93E-03 | 2.49E-03 | 1.06E-02 | 3.77E-03 | 2.78E-03 | 1.24E-03 | 1.15E-02 | 4.00E-03 | 0.43 | 0.21 |
| | 0.5hr | GWN | 9.04E-03 | 5.16E-03 | 3.20E-02 | 5.86E-03 | 4.00E-02 | 6.78E-03 | 4.87E-02 | 2.38E-02 | 2.23E-02 | 1.34E-02 | 1.25E-02 | 4.77E-03 | 9.86E-03 | 3.79E-03 | 2.51E-03 | 1.08E-03 | 9.87E-03 | 3.89E-03 | 9.31 | 2.17 |
| | 25hr | RW | 1.02E-02 | 6.68E-03 | 2.72E-02 | 7.82E-03 | 3.20E-02 | 9.57E-03 | 9.71E-03 | 6.29E-04 | 2.83E-02 | 1.33E-02 | 8.37E-03 | 1.35E-02 | 1.09E-02 | 4.04E-03 | 2.88E-03 | 1.25E-03 | 1.15E-02 | 5.17E-03 | 0.48 | 0.23 |
| | 25hr | GW | 8.83E-03 | 5.54E-03 | 2.71E-02 | 5.03E-03 | 4.27E-02 | 8.26E-03 | 6.03E-02 | 1.42E-02 | 1.92E-02 | 1.57E-02 | 1.15E-02 | 2.97E-03 | 1.09E-02 | 4.25E-03 | 2.72E-03 | 1.14E-03 | 1.06E-02 | 4.88E-03 | 7.46 | 2.23 |
| | 25hr | GWN | 1.27E-02 | 1.07E-02 | 2.33E-02 | 3.16E-03 | 3.40E-02 | 6.20E-03 | 1.31E-02 | 2.14E-03 | 2.52E-02 | 7.59E-03 | 1.25E-02 | 3.88E-03 | 1.19E-02 | 5.03E-03 | 2.95E-03 | 1.22E-03 | 1.23E-02 | 6.43E-03 | 0.41 | 0.18 |
| 398 days | 0.5hr | RW | 1.17E-02 | 5.60E-03 | 7.19E-03 | 3.68E-03 | 1.55E-02 | 1.03E-02 | 8.15E-02 | 4.87E-02 | 1.02E-02 | 5.98E-03 | 9.52E-03 | 3.91E-03 | 8.69E-03 | 3.71E-03 | 3.24E-03 | 3.09E-03 | 2.55E-02 | 1.57E-02 | 11.27 | 7.87 |
| | 0.5hr | GW | 1.24E-02 | 5.93E-03 | 7.53E-03 | 1.07E-03 | 1.31E-02 | 3.99E-03 | 4.73E-02 | 4.14E-02 | 1.53E-02 | 8.77E-03 | 7.79E-03 | 1.93E-03 | 7.87E-03 | 2.15E-03 | 1.99E-03 | 5.41E-04 | 2.62E-02 | 1.21E-02 | 0.31 | 0.08 |
| | 0.5hr | GWN | 9.63E-03 | 3.36E-03 | 3.73E-02 | 1.26E-02 | 4.49E-02 | 1.70E-02 | 1.36E-01 | 1.04E-01 | 7.78E-03 | 2.99E-02 | 1.04E-02 | 1.99E-03 | 8.98E-03 | 2.52E-03 | 2.97E-03 | 1.49E-03 | 2.74E-02 | 1.35E-02 | 7.66 | 1.96 |
| | 25hr | RW | 1.25E-02 | 4.81E-03 | 3.35E-02 | 8.60E-03 | 3.78E-02 | 1.20E-02 | 4.70E-02 | 4.93E-02 | 1.40E-02 | 4.61E-03 | 1.22E-02 | 3.19E-03 | 1.20E-02 | 2.19E-03 | 3.98E-03 | 1.62E-03 | 3.02E-02 | 9.84E-03 | 0.37 | 0.12 |
| | 25hr | GW | 1.04E-02 | 3.28E-03 | 3.18E-02 | 1.13E-02 | 4.81E-02 | 1.94E-02 | 1.59E-01 | 1.04E-01 | 5.94E-03 | 2.25E-03 | 1.05E-02 | 2.66E-03 | 2.07E-02 | 1.99E-02 | 2.34E-03 | 5.94E-04 | 2.60E-02 | 1.15E-02 | 8.65 | 0.87 |
| | 25hr | GWN | 1.06E-02 | 4.46E-03 | 2.66E-02 | 9.63E-03 | 3.48E-02 | 1.74E-02 | 7.45E-02 | 8.50E-02 | 1.32E-02 | 7.21E-03 | 9.20E-03 | 8.08E-04 | 9.51E-03 | 1.64E-03 | 2.36E-03 | 4.33E-04 | 2.52E-02 | 1.05E-02 | 0.46 | 0.16 |